# The Association between Spicy Food Consumption and Psychological Health in Chinese College Students: A Cross-Sectional Study

**DOI:** 10.3390/nu14214508

**Published:** 2022-10-26

**Authors:** Chunchao Zhang, Wenhao Ma, Zhiqing Chen, Chaoqun He, Yuan Zhang, Qian Tao

**Affiliations:** 1Department of Public Health and Preventive Medicine, School of Basic Medicine and Public Health, Jinan University, Guangzhou 510632, China; 2Guangdong-Hongkong-Macau Institute of CNS Regeneration, Ministry of Education CNS Regeneration Collaborative Joint Laboratory, Jinan University, Guangzhou 510632, China; 3Division of Medical Psychology and Behavior Science, School of Basic Medicine and Public Health, Jinan University, Guangzhou 510632, China

**Keywords:** psychological health, college students, spicy food, chili peppers

## Abstract

Background: Capsaicin is the main active ingredient in chili peppers and spicy food. Animal studies provide contradictory results on the role of capsaicin in psychiatric disorders. There are no epidemiological studies to investigate the relationship between spicy food consumption and psychological health. Methods: A cross-sectional online survey was conducted. Psychological health was assessed with the Depression Anxiety Stress Scale, and spicy food consumption was assessed as frequency, strength, and duration of consumption. Multivariable logistic regression was conducted to determine the associations between spicy food consumption and psychological symptoms. Results: Our sample comprised 1771 participants (male = 674, mean age = 21.97 years). The odds of having depressive, anxiety, and stress symptoms were 34.0%, 46.5%, and 19.1% in Chinese college students, respectively. After adjusting for a series of covariates, compared with non-consumers, the odds ratios (ORs) of depressive symptoms across spicy food consumption were 1.13 (95% CI: 0.87–1.46) for 1–2 days/week and 1.38 (95% CI: 1.02–1.86) for ≥3 days/week. With regard to anxiety symptoms, the ORs were 0.99 (95% CI: 0.78, 1.27) for 1–2 days/week and 1.50 (95% CI: 1.13–1.99) for ≥3 days/week. For stress symptoms, the ORs were 0.90 (95% CI: 0.66–1.23) for 1–2 days/week and 1.27 (95% CI: 0.89–1.80) for ≥3 days/week. The ORs for the depressive symptoms associated with different intensities of spicy food consumption were 1.00 (reference) for the reference group (non-consumers), 1.17 (95% CI: 0.90–1.52) for eating weakly spicy food, and 1.34 (95% CI: 1.01–1.78) for moderately to strongly spicy food. Conclusion: The findings suggested a positive association between frequently spicy food consumption and depressive/anxiety symptoms in adolescents, and no such association was found for stress symptoms.

## 1. Introduction

Adolescent psychological well-being is a public health priority, with mental disorders affecting 10% to 20% of adolescents worldwide [1]. College students are at an important stage of overall physical and mental development, and they face a variety of challenges as they struggle with dealing with changes in bodies and hormones due to puberty, living on their own, being away from home for the first time, increasing academic stress, and making important decisions. Increasing research suggests that the prevalence of depression, anxiety, and stress is higher among college students compared with the general population, and this prevalence seems to be increasing [2]. Poor mental health in college students has far-reaching effects on every aspect of life, such as impaired academic performance, disturbed sleep, impaired social functioning, and increased substance abuse [3]. These in turn create long-term morbidity and a substantial burden on society [4]. At their worst, mental disorders, particularly depression, can lead to self-harm and suicide, which is the fourth leading cause of death in adolescents [5]. The recognition of controllable risk factors in life is, therefore, crucial for preventing mental illnesses and reducing their prevalence.

Accumulating evidence has confirmed a notable relationship between a better-quality diet and better psychological health outcomes in the adult population [6] as well as in children and adolescents [7]. For instance, some epidemiological research found that fruit and vegetable intake was protectively associated with mental health [8]. A recent study further indicated banana consumption had a link to depressive symptoms in the Chinese general adult population [9]. Mental health has also been related to drinks, such as sugar-sweetened beverages and coffee [10,11]. In contrast, a poor diet, such as increased consumption of junk foods, is related to the risk of depression and anxiety [12]. Chili peppers, as a common seasoning and flavoring spice, have been ubiquitously used in spicy cuisines and are largely present in different cultures’ diets worldwide [13], such as the Hispanic diet in Mexico and Latin America and the Chuan diet in China. In many parts of China, such as Sichuan and Hunan, more than 30% of adults consume chili and spicy food daily [14]. Chili peppers are rich in minerals, vitamins, and amino acids that are essential for human health [15]. However, several epidemiological studies suggested that chili and spicy food were positively related to obesity [16], hyperuricemia [17], self-reported fractures [18], junk food consumption, tobacco smoking [19], and alcohol drinking [20,21]. A study also demonstrated a positive association between high chili intake and cognitive decline in a Chinese population-based cohort study [14]. 

Despite this research, the chili–mental health relationship is largely unknown. It has been hypothesized that capsaicin, the active ingredient in chili peppers, is responsible for their effects. A vanilloid receptor known as TRPV1 (transient receptor potential vanilloid subtype 1) is the receptor for capsaicin and is universally distributed in the sensory nerves, brain, and blood vessels [15]. Animal studies have shown that chili pepper intake may be associated with mental health, such as anxiety, depression, and stress [22]. The activation of TRPV1 by the administration of agonists such as capsaicin and resinoxin to rodents elicits anxiety responses and depression-related behaviors, while antagonists induce anxiolytic and antidepressant-like effects in rodents [23,24,25]. Preclinical studies show that capsaicin promotes long-term potentiation (LTP) by activating TRPV1 and inhibits long-term depression (LTD) in the hippocampus, resulting in an anti-stress effect [26]. However, there is less research on chili intake and human emotions.

The current cross-sectional study aims to investigate the relationship between spicy food consumption and psychological health in Chinese college students. To the best of our knowledge, this is the first epidemiological study to investigate the connection between spicy food consumption and the odds of having depressive, anxiety, and stress symptoms. It was hypothesized that high consumption of chilies and spicy food was positively associated with the occurrence of symptoms of depressive and anxiety and negatively associated with stress.

## 2. Methods

### 2.1. Participants 

Based on our previous survey, it was estimated that the odds of having depressive, anxiety, and stress symptoms among college students in China were 24.0%, 38.8%, and 21.1%, respectively [27]. According to the cross-sectional survey sample size calculation formula, we aimed to enroll at least 1581 participants considering anticipated attrition and missing data of 10%. A cross-sectional survey was conducted among college students in Guangdong Province, China, using a structured questionnaire. The data were collected from February 2022 to July 2022. Snowball sampling was adopted to distribute online questionnaires to university students using the Wenjuanxing online survey platform through the WeChat group and WhatsApp messaging application. This research was approved by the ethics committee of Jinan University, and participants provided electronic written consent before answering the questions. 

### 2.2. Evaluation of General Information

Demographic information was collected, including age, gender, grade, height, weight, education level, major, and monthly living expenses. Body mass index (BMI) was calculated by dividing the weight (kg) by the square of the height (m^2^). In addition, information related to lifestyles, such as alcohol consumption (never, currently drinking), tobacco smoking (never, currently smoking), sleep quality (low, medium, high), and physical activity (≥2 days/week or <2 days/week) was collected. Other information related to personal physical health (cancer, heart attack, and other severe organic disease), self-reported history of mental illness (yes, no), and family history of mental illness (yes, no) was also collected.

### 2.3. Evaluation of Psychological Health

Psychological health was assessed with the 21-item Depression Anxiety Stress Scale (DASS-21) [28]. The DASS-21 is a mature tool for evaluating psychological symptoms. The full scale contains 21 items, and the three subscales of depression, anxiety and stress each contain 7 items, all of which are scored on a 4-point scale from 0 (inconsistent) to 3 (always consistent). Multiply the scores of each subscale by 2, which is the score of the subscale. The higher the score, the more serious the symptom is. If the scores on the three subscales of depression, anxiety, and stress are higher than 10, 8, and 16, the subject is defined as suffering from depressive, anxiety, and stress symptoms, respectively. The Chinese version of the DASS-21 was used in this study, and this scale has been verified among Chinese hospital staff and university students [29,30]. Among university students in China, Cronbach’s alphas for the subscales of depression, anxiety, and stress were 0.83, 0.80, and 0.82, respectively. Our sample also demonstrated excellent reliability, with Cronbach alphas of 0.860, 0.804, and 0.840 for these three subscales, respectively, and 0.937 for the total scale of DASS-21. 

### 2.4. Evaluation of Spicy Food Consumption

Spicy food consumption was assessed with four items that had been validated in a previous study and had been widely used in epidemiology studies [13,31]. We asked the participants “Do you eat spicy food?” (yes, no). The participant responding with a “yes” would proceed to the next question: “During the past month, how often did you eat spicy food?” (never/almost never, only occasionally, 1–2 days per week, 3–5 days per week, or 6–7 days per week). The participants who consumed spicy food at least once per week were defined as regular consumers, and they were asked to provide additional information about their preferred spice intensity (weak, moderate, or strong) and years eating spicy food (< 5 years, 5–10 years, more than 10 years, or uncertain). 

### 2.5. Statistical Analysis

Questionnaires with more than 10% missing data were considered invalid. The expectation–maximization algorithm was applied to deal with missing data on the questionnaires [32]. The descriptive data of study participants were calculated using the arithmetical means (95% confidence interval, CI) or the appropriate percentage. The analysis of variance (ANOVA) was used for normal continuous variables, the Kruskal-Wallis test was used for continuous variables without normal distribution or homogeneity of variance, and the chi-square test was used for classified variables. For further analysis, the consumption of spicy food (frequency, intensity, and length) was used as an independent variable, and depression, anxiety, and stress symptoms were used as dependent variables. To test the associations between spicy food consumption and psychological health, multivariate logistic regression models were used for calculating odds ratios (ORs) and confidence intervals (95% CIs), with the lowest category (frequency: never or almost never; strength: never or almost never; duration: never or almost never) treated as the reference group. A set of models was used with considerations of different covariates: Model 1 was a crude model with no covariates; model 2 adjusted for demographic covariates of gender, age, education, monthly living expenses, and BMI; model 3 further adjusted for alcohol consumption and tobacco smoking; model 4 further adjusted for physical activity, sleeping quality, and the family history of mental health. All statistical analyses were performed using SPSS software, version 21.0 (IBM, Armonk, NY, USA). All statistical tests were bilateral, and *p* value < 0.05 was considered to be statistically significant. 

## 3. Results

### 3.1. Descriptive Results

A total of 1867 participants participated in this questionnaire. The participants with the following conditions were excluded for analysis: >10% missing data or impaired physical/mental health. As a result, data for 1717 participants were analyzed. Due to the limited number of individuals in the categories of occasionally eating spicy food (8 cases) and eating 6–7 days per week (114 cases), the 2 categories were merged with the existing categories. Overall, 26.4% of participants reported never or almost never consuming spicy food, 48.3% reported consuming 1–2 days per week, and 25.3% reported consuming ≥3 days per week. The odds of having depressive, anxiety, and stress symptoms were 34.0%, 46.5%, and 19.1%, respectively. The participants’ characteristics according to frequencies of spicy food consumption are shown in Table 1.

### 3.2. Frequency of Spicy Food Consumption

The frequency of spicy food consumption was positively related to depressive and anxiety symptoms (Table 2). In the fully adjusted model (model 4), compared with the reference group (non-consumers), those who consumed spicy food ≥3 days per week had higher odds of having depressive symptoms (OR = 1.38, 95%CI (1.02–1.86)) and anxiety symptoms (OR = 1.50, 95%CI (1.13–1.99)). No statistical significance was found between the frequency of spicy food intake and the occurrence of stress symptoms. 

### 3.3. Strength of Spicy Food Consumption

The strength of spicy food consumption was positively related to depressive symptoms (Table 3). Overall, compared with the reference group (non-consumers), higher intensity of spicy food intake was associated with higher odds of having depressive symptoms (OR = 1.34, 95% CI (1.01–1.78)). Spicy food intensity was not found to be associated with anxiety or stress symptoms.

### 3.4. Duration of Spicy Food Consumption

The duration of spicy food consumption was weakly positively related to depressive and anxiety symptoms (Table 4). In a few models, the duration of spicy food consumption appeared to be positively associated with symptoms of depression and anxiety, such as in Model 2. However, when further covariates were included in the model, no similar correlation was found. Overall, the duration of spicy food intake has little or no relationship with the development of psychiatric symptoms.

## 4. Discussion

To the best of our knowledge, we are the first to investigate the correlations between spicy food intake and psychological health in Chinese college students. Our results indicated a positive correlation between spicy food consumption and the occurrence of depressive and anxiety symptoms but not for stress symptoms. Specifically, individuals who consumed spicy food three or more times per week showed an increased possibility of developing symptoms of depression and anxiety. Moreover, individuals who ate moderately to heavily spicy food were more likely to have depressive symptoms, compared with those who did not have any spicy food in their daily diet. No such relationship was observed between spicy food consumption and anxiety/stress symptoms. Furthermore, years of spicy food consumption did not affect the odds of having psychological symptoms. 

A series of multiple confounding factors was adjusted in this study. Firstly, demographic factors, such as age [33], gender [34], education [35], monthly living expenses [36], and BMI [37], have been shown to relate to psychological health in college students, and thus, we adjusted these five variables in model 2. The adjustment did not change the association between spicy food consumption and depressive, anxiety, and stress symptoms. Secondly, lifestyle factors, such as alcohol drinking and smoking, may relate to psychological health [38,39], which were further adjusted in model 3. Similar associations still existed between spicy food consumption and depressive, anxiety, and stress symptoms. The moderate-to-heavy spicy group had a higher risk of depressive symptoms compared with the non-consumers group, although the difference was not statistically significant (*p* = 0.06). Finally, physical activity [40], sleep quality [41], and family history of mental disorders [42] may relate to mental health, and were added in model 4. These results were almost unchanged when adjusted for these confounding factors.

Spicy food is featured in the Hispanic culture diet in Mexico and the Asian diet cultures in China [43]. Capsaicin is the main active ingredient in chili peppers and spicy food [15]. Some studies have shown that capsaicin is neurotoxic [44,45], and high spicy food intake was positively correlated with worse cognitive scores and self-reported poor memory [14]. This might explain why no consumption and lower levels of spicy food intake were not associated with psychiatric symptoms, but higher levels were positively associated with depressive and anxiety symptoms. Our research found that the duration of spicy food intake was not correlated with the odds of having psychological symptoms, which may be due to the desensitization effect of long-term exposure [46].

However, the mechanisms linking spicy food and psychological symptoms have yet to be fully elucidated. Health benefits of spicy food have been ascribed to capsaicin, the main active ingredient in chili peppers. Evidence from animal studies appears to provide conflicting results on capsaicin’s role in mood [47]. Transient receptor potential vanilloid subtype 1 (TRPV1), the receptor for capsaicin, is widely distributed in humans [15]. The activation of the TRPV1 channel in mice has been shown to affect mood [48]. It is evidenced that TRPV1 knockout mice (TRPV1^(-/-)^) exhibit fewer anxiety-related behaviors on the light–dark test and elevated plus maze compared with wild-type littermates [49]. Capsaicin and olvanil (a capsaicin analog) have been reported to activate TRPV1, thereby exerting antidepressant-like effects that were correlated with altered expressions of GABAA, 5-HT1A, and NMDA receptors in mice [50,51]. However, the activation of TRPV1 channels results in an increase in the release of glutamate from the synaptic cleft and a decrease in the release of GABA and dopamine (DA) [52]. Additionally, capsaicin upregulates histone deacetylase 2 (HDAC2) via TRPV1 and impairs neuronal maturation and synaptic plasticity in the hippocampus in rodents, affecting cognitive and emotional functions [53]. This may explain the activation of TRPV1 by agonists such as capsaicin and resinotoxins to induce anxiety responses and depression-related behaviors in rodents. TRPV channels have also been associated with neuropsychiatric disorders, including stress, depression, and anxiety [22,47]. In summary, the effect of capsaicin-activated TRPV1 on neuropsychiatric disorders in animal models remains unclear and needs further investigation.

Compared with non-consumers of spicy food, participants with the higher consumption of spicy food showed a higher body mass index (BMI). Numerous previous studies can support our research findings. A report from China Kadoorie Biobank found that the intensity and frequency of spicy foods may be related to increased BMI and other measures of obesity [16]. A survey from China’s Henan Province showed that spicy taste and frequency of spicy food are correlated with a higher risk of general obesity in the rural population [54]. Obesity is a key risk factor for anxiety and depression; elevated BMI predicts a chronic course of anxiety and depressive symptoms [55]. Obese adults have increased odds of depressive and anxiety symptoms compared with non-obese people [56,57]. Meanwhile, our results found that current alcohol drinkers consumed spicy food more frequently compared with non-drinkers. This is consistent with previous research findings. For example, a study from South Korea found that a higher preference for spicy food was possibly one risk factor for alcohol dependence [21]. Some studies have shown that consumers of more frequent or spicy foods are more likely to be drinkers [20,58]. In addition, alcohol is one of the most commonly abused drugs and central nervous system (CNS) depressants. High alcohol consumption interferes with brain communication pathways linked to psychological symptoms [59]. Several previous studies have found that increased exposure to alcohol can mediate the occurrence of depression and anxiety [60,61]. According to the above results, it can be inferred that more frequent and stronger spicy food consumption may enhance energy and alcohol intake, leading to higher odds of depressive and anxiety symptoms.

Several limitations should be considered in our present study. First, we used a cross-sectional study design, which precluded the ability to establish causal inferences. In future work, we hope to conduct follow-up or clinical studies in adolescents to determine the causal role of spicy food consumption in the development of depressive and anxiety symptoms. Second, we were unable to explore its underlying mechanisms due to the shortage of human-related biomarkers. Third, our sample was limited to college students in Guangdong Province, which may not good enough to generalize to the whole population. Future researchers may wish to replicate the results in other populations.

## 5. Conclusions

In conclusion, the consumption of spicy food is associated with depressive and anxiety symptoms, but not with stress symptoms, in a Chinese college student population. The findings suggested a positive association between frequent spicy food consumption and depressive/anxiety symptoms in adolescents. A similar association was observed between moderately to strongly spicy food consumption and depressive symptoms.

## Figures and Tables

**Table 1 nutrients-14-04508-t001:** Characteristics of the study subjects according to categories of spicy food consumption.

	Spicy Food Consumption			
Characteristics	Never or Almost Never (*n* = 453)	1 or 2 Days/Week(*n* = 830)	≥3 Days/Week(*n* = 434)	*p* ^a^
Males (%)	48.8	38.4	30.9	<0.001
Depressive symptoms (%)	30.2	34.1	37.6	0.071
Anxiety symptoms (%)	43.5	45.1	52.3	0.016
Stress symptoms (%)	18.3	18.0	22.1	0.178
Age (y) ^b^	21.9 ± 2.4	22.0 ± 2.5	22.2 ± 2.4	0.213
BMI ^c^	20.9 ± 3.5	21.5 ± 3.9	21.1 ± 3.5	0.020
Do you have siblings? (%)				
Yes	22.5	16.6	21.2	0.020
No	77.5	83.4	78.8	
Monthly disposable income (%)				
≤1000	17.7	15.1	11.8	0.010
1000–2000	65.6	68.4	65.2	
>2000	16.8	16.5	23.0	
Education level (%)				
Junior college	7.5	8.1	7.4	< 0.001
College/university	71.7	71.4	56.0	
Graduate school	20.8	20.5	36.6	
Physical activity (%)				
≥2 days/week	43.3	39.6	35.0	0.042
<2 days/week	56.7	60.4	65.0	
Smoking (%)				
Never	96.0	95.4	95.6	0.880
Current	4.0	4.6	4.4	
Alcohol use (%)				
Never	66.4	56.4	55.5	0.001
Current	33.6	43.6	44.5	
Sleep quality (%)				
High	54.7	53.3	52.3	0.604
Medium	35.3	34.2	36.9	
Low	9.9	12.5	10.8	
Strength of spice (%)				
Weak	-	79.6	48.8	<0.001
Moderate	-	18.4	41.5	
Strong	-	2.0	9.5	
Duration of spicy food consumption (%)				
<5 years	-	43.5	19.1	<0.001
5–10 years	-	24.5	21.4	
>10 years	-	32.0	59.4	

^a^ Determined by ANOVA, chi-square, or Kruskal–Wallis test. ^b^ All values are X ± SD. ^c^ Body mass index. Calculated from participants’ measured weight and height.

**Table 2 nutrients-14-04508-t002:** Regression coefficients (95% CI) for psychological health by frequency of spicy food consumption among Chinese college students.

Logistic Regression Models	Frequency of Spicy Food Consumption								
	Depressive Symptom			Anxiety Symptom			Stress Symptom		
	Never or Almost Never	1–2 Days/Week	≥3 Days/Week	Never or Almost Never	1–2 Days/Week	≥3 Days/Week	Never or Almost Never	1–2 Days/Week	≥3 Days/Week
Model 1	1.00 (Ref.)	1.19 (0.93, 1.53) ^a^	1.39 (1.05, 1.83)	1.00 (Ref.)	1.07 (0.85, 1.34)	1.43 (1.09, 1.86)	1.00 (Ref.)	0.98 (0.73, 1.31)	1.27 (0.91, 1.76)
Model 2	1.00 (Ref.)	1.23 (0.95, 1.57)	1.51 (1.13, 2.01)	1.00 (Ref.)	1.07 (0.85, 1.36)	1.61 (1.22, 2.10)	1.00 (Ref.)	0.99 (0.73, 1.33)	1.39 (0.99, 1.95)
Model 3	1.00 (Ref.)	1.16 (0.90, 1.49)	1.39 (1.04, 1.86)	1.00 (Ref.)	1.02 (0.81, 1.30)	1.50 (1.14, 1.98)	1.00 (Ref.)	0.95 (0.70, 1.29)	1.31 (0.93, 1.85)
Model 4	1.00 (Ref.)	1.13 (0.87, 1.46)	1.38 (1.02, 1.86)	1.00 (Ref.)	0.99 (0.78, 1.27)	1.50 (1.13, 1.99)	1.00 (Ref.)	0.90 (0.66, 1.23)	1.27 (0.89, 1.80)

Model 1: crude model. Model 2: adjusted for sex, age education, monthly living expenses, and BMI. Model 3: adjusted for Model 2 and covariates of alcohol consumption and tobacco smoking. Model 4: The whole model, adjusted for Model 3 and covariates of physical activity, sleep quality, and family history of mental health. ^a^ Odds ratio (95% confidence interval) (all such values).

**Table 3 nutrients-14-04508-t003:** Regression coefficients (95% CI) for psychological health by the strength of spice among Chinese college students.

Logistic Regression Models	Strength of Spicy Food Consumption								
	Depressive Symptoms			Anxiety Symptoms			Stress Symptoms		
	Never	Weak	Moderate to Strong	Never	Weak	Moderate to Strong	Never	Weak	Moderate to Strong
Model 1	1.00 (Ref.)	1.25 (0.98, 1.60) ^a^	1.35 (1.01, 1.80)	1.00 (Ref.)	1.20 (0.95, 1.51)	1.20 (0.91, 1.57)	1.00 (Ref.)	1.00 (0.74, 1.34)	1.20 (0.86, 1.68)
Model 2	1.00 (Ref.)	1.30 (1.01, 1.67)	1.42 (1.06, 1.91)	1.00 (Ref.)	1.23 (0.97, 1.56)	1.27 (0.96, 1.67)	1.00 (Ref.)	1.02 (0.76, 1.38)	1.22 (0.90, 1.79)
Model 3	1.00 (Ref.)	1.20 (0.93, 1.55)	1.33 (0.99, 1.79)	1.00 (Ref.)	1.16 (0.91, 1.47)	1.21 (0.91, 1.60)	1.00 (Ref.)	0.95 (0.71, 1.30)	1.21 (0.86, 1.72)
Model 4	1.00 (Ref.)	1.17 (0.90, 1.52)	1.34 (1.01, 1.78)	1.00 (Ref.)	1.13 (0.89, 1.44)	1.19 (0.89, 1.58)	1.00 (Ref.)	0.90 (0.66, 1.23)	1.17 (0.82, 1.68)

Model 1: crude model. Model 2: adjusted for sex, age education, monthly living expenses, and BMI. Model 3: adjusted for Model 2 and covariates of alcohol consumption and tobacco smoking. Model 4: The whole model, adjusted for Model 3 and covariates of physical activity, sleep quality, and family history of mental health. ^a^ Odds ratio (95% confidence interval) (all such values).

**Table 4 nutrients-14-04508-t004:** Regression coefficients (95% CI) for psychological health by the duration of spicy food consumption among Chinese college students.

Logistic Regression Models	Duration of Spicy Food Consumption											
	Depressive Symptoms				Anxiety Symptoms				Stress Symptoms			
	Never	<5 Years	5–10 Years	>10 Years	Never	<5 Years	5–10 Years	>10 Years	Never	<5 Years	5–10 Years	>10 Years
Model 1	1.00 (Ref.)	1.33 (1.00, 1.76) ^a^	1.27 (0.93, 1.74)	1.26 (0.96, 1.65)	1.00 (Ref.)	1.22 (0.94, 1.59)	1.06 (0.79, 1.40)	1.26 (0.98, 1.63)	1.00 (Ref.)	0.98 (0.65, 1.43)	0.90 (0.61, 1.33)	1.23 (0.89, 1.69)
Model 2	1.00 (Ref.)	1.34 (1.01, 1.78)	1.34 (0.97, 1.84)	1.34 (1.01, 1.77)	1.00 (Ref.)	1.20 (0.91, 1.56)	1.08 (0.80, 1.46)	1.40 (1.08, 1.82)	1.00 (Ref.)	0.98 (0.69, 1.38)	0.94 (0.63, 1.38)	1.32 (0.95, 1.83)
Model 3	1.00 (Ref.)	1.14 (0.80, 1.64)	1.22 (0.88, 1.68)	1.24 (0.94, 1.65)	1.00 (Ref.)	0.87 (0.61, 1.22)	0.99 (0.73, 1.35)	1.32 (1.01, 1.72)	1.00 (Ref.)	0.82 (0.53, 1.29)	0.88 (0.59, 1.31)	1.26 (0.91, 1.76)
Model 4	1.00 (Ref.)	1.18 (0.81, 1.70)	1.32 (0.95, 1.83)	1.18 (0.88, 1.57)	1.00 (Ref.)	0.88 (0.62, 1.25)	1.05 (0.77, 1.44)	1.26 (0.96, 1.66)	1.00 (Ref.)	0.82 (0.52, 1.26)	0.94 (0.63, 1.43)	1.17 (0.83, 1.64)

Model 1: crude model. Model 2: adjusted for sex, age education, monthly living expenses, and BMI. Model 3: adjusted for Model 2 and covariates of alcohol consumption and tobacco smoking. Model 4: The whole model, adjusted for Model 3 and covariates of physical activity, sleep quality, and family history of mental health. ^a^ Odds ratio (95% confidence interval) (all such values).

## Data Availability

Data can be received by asking the corresponding author.

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
