# Peer review of "The Association between Spicy Food Consumption and Psychological Health in Chinese College Students: A Cross-Sectional Study"

_nutrients, 2022, doi:10.3390/nu14214508_

Round 1

Reviewer 1 Report

The work presented is very well done.

However, in the future, I recommend the authors to choose more appropriate research methods.

In my opinion, the questionnaire used distorted the results. Although the initial intention was very well designed.

Why did you choose an anonymous questionnaire? Wouldn't it be more appropriate to contact university students by phone with the help of a nutritionist, a sensory expert?

Author Response

Dear reviewer:

We greatly appreciate the chance of revising this manuscript. The comments are very useful which further improve the content of the paper. Attached is our detailed responses to comments of the reviewer 1.

Reviewer 2 Report

This is an intriguing study on the association between spicy food in take and mood.  The text is well written, and the study was well conducted.  The statistics were generally appropriate with a couple comments.  I have modest suggestions for some simplifications and a few additions.

Abstract

Lines 27-29: These values represent the univariate ORs.  Usually, authors present the fully adjusted ORs having considered them the best model of the situation.  Please present the ORs from model 5 here. Also missing here are the model 5 results for stress.  It is equally interesting that there is no association found for spicy food and stress, especially given the original hypothesis.

Lines 33-34: A fuller Conclusion would be that a positive association was found for depression and anxiety but not for stress.

Introduction

Line 63: The Mediterranean diet is not well known for their consumption of hot spicy foods as much as diets from Hispanic cultures such as from Mexico and Latin America or from southeast Asia.

Line 70 or 71: A paragraph break might be appropriate here for a change in topic.

Line 74: The English in the paper is excellent overall.  I suggest one more careful reading to catch small errors such as “humans body” in this line.

Results

Line 166: The number of significant digits reported in a result usually depends on the amount of variability usually as expressed in the standard deviation.  In this case, with a standard deviation of 2.47, there is little value in expressing a mean beyond a whole number such as 22.  However, 22.0 with SD=2.5 is probably also acceptable.  This rule can easily be applied elsewhere.

Table1

There is clearly an issue with this table formatting that I will let the authors and publishers handle.

Why do we need the DASS designations here.  It only adds confusion to the table.  Can it just be Depressive symptoms (%), Anxiety symptoms (%) and Stress symptoms (%).

Lines 185-196: It is not necessary to repeat in text what is already reported in a table.  One can simply summarize the table briefly.  The section should be reduced substantially.

Table 2.

This is a very crowded table.  I suggest possibly removing the ref. columns.  The two rows with the numbers of students are not absolutely necessary and often these absolute values are not reported in epi papers.  It is not clear in the footnotes that the models are additive, that the named factors are added to the previous model.  Also, unless there is substantial insight gained by all five models, I might suggest 3-4 models.

Lines 204-218: Again, most of this information is a repeat of what is reported in Table 3.  This is not necessary. One can simply summarize the table briefly.  The section should be reduced substantially.

Table 3.

Similar suggestion for congestion as in Table 2. 

Lines 226-238: Similar suggestions to Lines 204-218.

Table 4.

Similar suggestions to Tables 2 and 3.  The extra width of Table 4 also suggests another solution to Tables 2 and 3 and that is to allow them to be wider, as with Table 4.

Discussion

Line 248: a positive association with “occurrence of depressive and anxiety symptoms” should be extended to include the null association with stress, per the authors’ original hypothesis.

Lines 272-278: Although the MD is easy to use as a comparison diet, it is a poor example of a spicy diet.  Other diets, such as from Latin America or from southeast Asia, are especially well known for their spiciness.  This leads to the rather confusing reference to the phytochemical in tea as examples from a spicy diet.  Are resveratrol and EGCG spicy?  This is puzzling.

Conclusion

Please mention the null association with stress.  Null findings are also important to communicate.

Author Response

Dear reviewer:

We greatly appreciate the chance of revising this manuscript.  Attached is our detailed responses to comments of the reviewer 2.
